# A Data Clustering Algorithm for Detecting Selective Forwarding Attack in Cluster-Based Wireless Sensor Networks

**DOI:** 10.3390/s20010023

**Published:** 2019-12-19

**Authors:** Hao Fu, Yinghong Liu, Zhe Dong, Yuanming Wu

**Affiliations:** School of Optoelectronic Science and Engineering, University of Electronic Science and Technology of China, Chengdu 610054, China; 201722050606@std.uestc.edu.cn (H.F.); liuyinghong@std.uestc.edu.cn (Y.L.); 201722050603@std.uestc.edu.cn (Z.D.)

**Keywords:** data clustering algorithm, selective forwarding attack, cumulative forwarding rate, cluster-based WSN

## Abstract

In cluster-based wireless sensor networks, cluster heads (CHs) gather and fuse data packets from sensor nodes; then, they forward fused packets to the sink node (SN). This helps wireless sensor networks balance energy effectively and efficiently to prolong their lifetime. However, cluster-based WSNs are vulnerable to selective forwarding attacks. Compromised CHs would become malicious and launch selective forwarding attacks in which they drop part of or all the packets from other nodes. In this paper, a data clustering algorithm (DCA) for detecting a selective forwarding attack (DCA-SF) is proposed. It can capture and isolate malicious CHs that have launched selective forwarding attacks by clustering their cumulative forwarding rates (CFRs). The DCA-SF algorithm has been strengthened by changing the DCA parameters (Eps, Minpts) adaptively. The simulation results show that the DCA-SF has a low missed detection rate of 1.04% and a false detection rate of 0.42% respectively with low energy consumption.

## 1. Introduction

A wireless sensor network (WSN) is a self-organizing network formed by a mass of small and cheap sensor nodes, which have low energy, poor computing ability, and small storage. The cluster-based WSN has been widely applied in large-scale data gathering WSNs [1,2]. In the dense cluster-based WSN, as shown in Figure 1, member nodes (MNs) send data packets to their cluster heads (CHs). Then, CHs forward these packets to the next-hop CHs until they reach the sink node (SN). In this way, each CH does not have to exchange data with the SN directly. On the one hand, the direct communication between the CH and SN may fail due to the long-distance or poor channels. On the other hand, in some conditions, the energy cost of multi-hop communication is less than that of direct long-distance communication. All the nodes in the network take turns to act as CHs, so the energy consumption can get balanced. The network lifetime also becomes longer.

CHs may usually suffer various attacks launched inside or outside, due to their vital roles in the cluster-based WSN. The selective forwarding attack is one of the most common attacks, where a CH compromised by the attacker drops all or part of the data packets. The severe influence includes not only high data loss and poor quality of service (QoS) but also the damage to energy-balanced routing protocols [3,4].

Selective forwarding attacks are divided into black hole attack, gray hole attack, and on/off attack. Black hole attack is the worst selective forwarding attack, since a malicious node drops all the packets it should forward [5]. In a gray hole attack, a malicious node randomly drops packets [6]. On/off attack is where a malicious node periodically drops all or part of data packets only within a time interval and acts normally in the other time [7]. In the selective forwarding attack, both the number and rounds of packets dropping are random, which is nearly impossible to be distinguished from that caused by poor channel quality [8]. This makes it easy for malicious nodes to hide their identities and increases the difficulty of detecting selective forwarding attacks. If a malicious node cannot be detected and isolated quickly, it would continue dropping data packets. Meanwhile, any node sentenced as a malicious node will be deprived of the rights owned by normal nodes, including participating in packet forwarding. The misdetection of normal nodes makes the number of data forwarding nodes in the network decrease, and the energy consumption of a single node increases. Consequently, the network lifetime will be shortened. In summary, under low energy consumption, the goals of detecting selective forwarding attacks [9] are as follows: low missed detection rate (MDR), low false detection rate (FDR), and high detection speed.

Most detection schemes on selective forwarding attacks employ neighbor monitoring nodes and their reputations without taking the impact of channel quality on detection into account [7,8,9,10,11,12,13]. Some detection schemes take advantage of the clustering algorithm [14], but the channel quality is not included in the data set collection process. This causes a normal node with a lower cumulative forwarding rate (CFR) to be misjudged as a malicious node due to a poor channel.

A poor channel affects the current forwarding rates and CFRs heavily. If the channel quality is ideal, the forwarding rate always maintains 100%; otherwise, the data packets have to be dropped. If the scheme does not consider the impact of channel quality on the CFR, it cannot distinguish who has made data packets dropped, malicious nodes or poor channels. So, it is difficult for a WSN to detect selective forwarding attacks. Most schemes assume the perfect channel and have a good performance, but in practice, they result in a low correct detection rate where a poor channel exists.

The motivation for utilizing the data clustering algorithm (DCA) is to find out anomalous CFRs of malicious nodes under the same channel with normal nodes. If a node is compromised and becomes malicious, its CFR will be anomalous to that of a normal node. Our scheme introduces one DCA based on density to distinguish the anomalous CFRs. It divides CFRs into different groups of similar ones. In the density-based clustering DCA, it doesn’t need to set the number of clusters in advance, and it can divide the CFRs into proper clusters depending on the distribution of the CFRs. The time and space complexity of a DCA can reduce to a low level so that it can be performed in sensor nodes.

In this paper, Section 2 reviews related works to help understand the proposal of our scheme. Section 3 elaborates on the proposed DCA for detecting a selective forwarding attack (DCA-SF) scheme. Section 4 conducts some simulation experiments, analyzes the results, and evaluates the advantages of DCA-SF. Section 5 gives conclusions and future work.

## 2. Related Work

### 2.1. Schemes against Selective Forwarding Attack

Since the selective forwarding attack was first proposed by Karlof [10], many schemes for detecting it have come out.

Semantic and Abhijit [11] proposed a selective forwarding attack detection scheme under the classical clustering algorithm, LEACH (Low Energy Adaptive Clustering Hierarchy). A counter is equipped in the base station. The counter checks whether the CH receives the data packets correctly and whether data packets reach the base station. A node is judged by its reliable neighbor nodes depending on selective time-variant flooding tests. The node will be charged as malicious once it fails the tests. In this scheme, the base station detects CHs. However, this scheme can only work in a small-scale cluster-based WSN. As the nodes increase, more data packets than ever would be sent to some nodes in the flooding test. Finally, too much test energy consumption itself can bring severe results to the whole network. In addition, ignoring the influence of the channel quality, this scheme often regards normal nodes as malicious under poor channel quality.

The watchdog [15] can monitor neighbor nodes so well that it is often available to detect selective forwarding attacks in WSNs. The node uses the watchdog to monitor neighbor nodes whether they forward its data packets or not. As shown in Figure 2, the watchdog supervisor deployed on node A detects whether neighbor nodes forward packets from A.

The watchdog mechanism may fail to work correctly when both B and C are malicious nodes. This time C may launch an attack, as shown in Figure 2. B forwards the data packet sent by A to C. In this case, A’s watchdog monitor will detect that B has forwarded the packets from A, and will not regard B as a malicious node. However, if C drops A’s packets sent by B, the watchdog mechanism can not detect this event [16]. Schemes proposed in the literatures [8,17] have improved the watchdog mechanism. These schemes assign inspector nodes and cooperative nodes to monitor the relay nodes. The scheme in [8] provides a novel network model that will also be adopted in our scheme. In a cluster, there are three types of nodes: MN, CH, and inspector node (IN). The IN supervises whether the CH forwards the data packets from MNs in the cluster. Each time the CH and IN are replaced, IN communicates with the neighbor IN to prevent the collusion attack shown in Figure 2. The reputation is also introduced when judging whether the CH is a malicious node or not. However, in the scheme of [8], if both the CH and IN become malicious, their MNs cannot inform other clusters of the message that “the CH and IN are malicious”. In this case, CH and IN continue to collude and launch attacks, causing packets of MNs in the cluster to fail to forward. The scheme [17] adds a cooperative node to monitor nodes that forward packets based on the watchdog mechanism. Unfortunately, the cooperative node may become malicious. The scheme in [12] improves the detection against such collusion attacks. When the MNs in one cluster find that the IN ignores the attack behavior of the CH, they will isolate the CH and the IN, and elect a new CH and IN. However, the channel quality problem is not considered in the scheme. Poor channels will lower the reputation of a normal CH, which increases the probability of misjudging a normal node as a malicious node.

In recent years, machine learning algorithms have been increasingly used in various research fields. If machine learning algorithms are applied in detecting attacks in WSNs, it will be good for detecting cunning attack patterns. However, the scheme has to take its time complexity and space complexity into account in WSNs because of the small memory space and the limited energy of sensor nodes. The scheme in [14] uses the DCA E-DBSCAN [18] and random forest algorithm [19] to detect network attacks of WSN. In the scheme, the data set organized by data from CHs to a SN is clustered by E-DBSCAN in SN. By clustering data, SN can pick up the anomalous data points to distinguish malicious nodes from normal ones. E-DBSCAN is easy to be used in WSN for detecting network attacks. It can distinguish the abnormal data from the normal data. Unfortunately, it cannot differentiate malicious CHs from normal CHs. In the random forest algorithm, the training set used for off-line training is the KDD Cup 99 data set [20] composed of nearly five million 41-dimensional vectors, which is a classical training set used for computer network attacks detection. However, gathering these vectors will be a big overhead for a WSN.

### 2.2. Data Clustering Algorithm (DCA)

The DCA, which is known as unsupervised classification, can divide a data set into many clusters according to their similarities. Several different DCAs have been proposed. CURE [21] and BIRCH [22] are based on hierarchy. CURE uses representative points as central points in the cluster. This can identify a variety of complex shapes and clusters of different sizes. However, in our scheme, the number of samples in the data set is relatively small, so it is difficult to find representative points. Furthermore, the complexity of CURE is O(m^2^) for low dimensions. This is too high to be performed in the sensor node. The basic idea of BIRCH is that the importance of each data point must be different so that different data samples can be treated differently. The BIRCH algorithm has a unique data structure called a clustering feature tree (CF-TREE). However, if the distribution cluster of the data set is not convex, the clustering effect is not good enough. Spectral clustering [23] based on graph theory is widely used in image analysis. However, how to automatically determine the number of clusters has become one of the key problems to be solved in spectral clustering. STING [24] is a grid-based multi-resolution clustering technique that divides a spatial region into rectangular cells. This technique has a fast processing speed but may reduce the quality and accuracy of the cluster.

K-means [25] and K-medoids [26] are based on partitioning. In K-means and K-medoids, clusters are groups of data that are characterized by a small distance to the cluster center [27]. K-means divides a data set into *K* clusters, and it make points in the same cluster have high similarity and low similarity in different clusters. The center of one cluster is the mean of all the points in the cluster. Every point in the data set must be divided into one cluster based on the shortest distance to the cluster center. The time complexity of K-means is O(m), where *m* is the number of points in the data set. This is an acceptable complexity in sensor nodes. However, the value of *K* needs to set in advance, and the choice of initial centers is random. These are two disadvantages of K-means. To overcome these two disadvantages, some methods are proposed. In the literature [28], the proposed method sets a *K* value and initial centers by rival penalized competitive learning (RPCL) [29]. However, the time complexity is O(m^2^), which is too large to be performed in sensor nodes. Kaufman et al. [30] proposed a heuristic method of estimating the local density of data points to pick up initial cluster centers of K-means. Furthermore, Dhillon et al. [31] recalculated cluster centers during the iteration to optimize clustering performance. However, the time complexity or space complexity is more than O(m^2^), so they are not suitable to be performed in sensor nodes. K-medoids is similar to K-means. The most significant difference is that the center of one cluster in K-medoids is a point rather than the mean of points in the cluster as in K-means. This improvement avoids the impact of isolated points on the mean. Traditional K-medoids is based on the partitioning around medoids (PAM) [32]. The time complexity of K-medoids is O(m^2^). Furthermore, it also needs to set the value of *K* in advance. So, this algorithm is not suitable to be performed in sensor nodes.

Density peaks clustering (DPC) [27] based on density is a recently proposed clustering algorithm published in *Science* magazine. This algorithm is based on two assumptions: that cluster centers are characterized by a higher density than their neighbors and by a larger distance from points with higher densities. It is performed clustering in terms of each data point’s *ρ* value and *δ* value. For the data point *i*, the *ρ* value and *δ* value of point *i* are shown in Equations (1) and (2).
(1)$$\rho_{i}=\underset{j\neq i}{\sum }\chi \left(d_{ij}-d_{c}\right)$$
(2)$$\delta_{i}=\underset{j:\rho_{j}>\rho_{i}}{\min}\left(d_{ij}\right)$$

The cutoff distance *d_c_* is a preset parameter, and its value varies in different methods. *d_ij_* is the Euclidean distance between point *i* and point *j*, and *χ*(*x*) = 1 when *x* < 0; otherwise *χ*(*x*) = 0. That means that *χ*(*d_ij_* − *d_c_*) = 1 when the distance *d_ij_* between two points *i* and *j* is smaller than the preset value of *d_c_*. Furthermore, $\rho_{i}$ is the number of points whose distances from point *i* are less than *d_c_*. In Equation (2), *δ_i_* is measured by calculating the minimum distance between point *i* and any other point with higher density. However, if the point *i* has the highest density, its *δ* value cannot be calculated by Equation (2). So, for the point *i* with the highest density, *δ_i_* is calculated by Equation (3).
(3)$$\delta_{i}=\underset{j}{\max}\left(d_{ij}\right)$$

SNN-DPC was proposed in the literature [33]. In SNN-DPC, the *δ_i_* of the point *i* with the highest density is defined by Equation (4). Although Equation (3) and Equation (4) are different, these two calculating methods all can confirm central points of clusters.
(4)$$\delta_{i}=\underset{i\neq j}{\max}\left(\delta_{j}\right)$$

For the Jain data set in which two crescent-shaped clusters of different densities are intertwined with each other, SNN-DPC can cluster the data sets of this type perfectly, but the clustering effect of DPC is not perfect. For the Path-based data set, the clustering effect of SNN-DPC is also better than DPC.

The time complexity and space complexity of DPC are both O(m^2^). Some studies in the literature [33,34,35] proposed the improved methods based on DPC, but the time complexity and space complexity limit these algorithms to be performed in sensor nodes. If the data set is small, the choice of *d_c_* impacts on the clustering result greatly.

DBSCAN [36] does have some drawbacks, as the literature [33] points out. The main one relates to the declining clustering quality of high-dimensional data and variable-density clusters. However, it does not matter in our scenarios, where the dimension of data is only two, and the difference in the clusters density is less. As for the difficulty in setting parameters—such as Eps and Minpt-- adaptive methods are employed in many recent papers, just as in our DCA-SF. Moreover, it is the permissive conditions and strong anti-noise capability [33] that make DBSCAN a proper solution in our scenarios. The DCA-SF that is proposed in the paper is a light detection scheme against selective forwarding attacks. The DCA-SF based on the DP-DBSCAN utilizes the clustering points of CFRs to detect attacks, which is different from E-DBSCAN. It achieves a low FDR and MDR with a low extra energy consumption of nodes.

## 3. Details of Scheme

DCA-SF assumes that the most dangerous attacks are from CHs on a WSN. This section will describe the scheme’s details including the network layout, the cooperation of different roles of nodes, the density-based DCA, and the implementation of DCA-SF.

### 3.1. Frame of Detection Mechanism

The frame of detection is shown in Figure 3. One cluster contains three types of nodes: the cluster head (CH), member nodes (MNs), and the inspector node (IN). The cluster radius is half of the node communication radius. This ensures that the CHs and INs of adjacent clusters are within each other’s communication range.

MNs in one cluster send data packets consisting of information of the environment to the CH in the cluster. Then, the CH passes data packets to the next-hop CH along the CH route in Figure 3, until the data reach the SN. The IN calculates the CFR of the CH and CFRs of MNs in its cluster (the calculation method will be given in Section 3.2), and then passes them along the IN route to the SN shown in Figure 3, with the terminal of SN. Once the clustering of the network finished, the nodes in each cluster are determined, and the MNs in each cluster take turns as CHs and INs according to the detection results and election rules. In the i-th cluster, the MN with the highest residual energy will become a new CH_i_ and the MN with the second-highest residual energy will become a new IN_i_.

### 3.2. Cooperation of Different Roles of Nodes

Four roles of nodes, MN, CH, IN, and SN exist in the WSN. As mentioned, MNs play the roles of the CH and IN in turn. Most nodes are MNs, and what they are supposed to do in the cluster-based WSN is merely to collect the information of the environment and send these data packets to their CHs. Next, CHs transfer the data packets they received to the SN. (Assume that there is no data fusion at CHs.)

INs do not forward any data packets during this process. Instead, *IN_i_* calculates the CFR of CH_i_ (*CFR_CH_i_*) and the CFR of MNs (*CFR_MN_j_*) in the i-th cluster. Equations (5)–(7) and Equations (8) and (9) tell the details.
(5)$$RECI\_CH_{i}\left(n\right)=\{\begin{array}{cc} RECI\_CH_{i}\left(n-1\right)+reci\_CH_{i}\left(n\right), & n>1\\ reci\_CH_{i}\left(n\right), & n=1 \end{array}$$
(6)$$FORW\_CH_{i}\left(n\right)=\{\begin{array}{cc} FORW\_CH_{i}\left(n-1\right)+forw\_CH_{i}\left(n\right), & n>1\\ forw\_CH_{i}\left(n\right), & n=1 \end{array}$$
(7)$$CFR\_CH_{i}\left(n\right)=\frac{FORW\_CH_{i}\left(n\right)}{RECI\_CH_{i}\left(n\right)}$$

*RECI_CH_i_*(*n*) and *FORW_CH_i_*(*n*) are the total numbers of packets received by *CH_i_* and that of packets successfully forwarded by *CH_i_* in the first *n* rounds, while the lowercase letters refer to the current round or the n-th round.
(8)$$FORW\_MN_{j}\left(n\right)=\{\begin{array}{cc} FORW\_MN_{j}\left(n-1\right)+forw\_MN_{j}\left(n\right), & n>1\\ forw\_MN_{j}\left(n\right), & n=1 \end{array}$$
(9)$$CFR\_MN_{j}\left(n\right)=\frac{FORW\_MN_{j}\left(n\right)}{n}$$
in data-gathering WSNs, every MN sends one data packet to its CHs in each round. Obviously, the *MN_j_* totally sends *n* data packets, of which *FORW_MN_j_*(*n*) of them reach the CH.

Only SNs and INs detect attacks, and MNs do not. Firstly, the SN screens the suspected CHs according to the CFR_CHs from the IN routes shown in Figure 3. The CHs charged as malicious may be innocent, and the misdetection is due to the poor channel at some local areas. Secondly, INs compare the CFRs of suspected CHs to those of their MNs, which are in the same channel condition with suspected CHs. Once the suspected CH keeps the same channel condition as its MNs in terms of the CFR, it will be released or it will be isolated.

### 3.3. DP-DBSCAN DCA

#### 3.3.1. Introduction of DBSCAN

DCA is a procedure of gathering data satisfying some criteria into one group. This technology has already been widely applied in various fields including pattern recognition and machine learning [36]. In DCA-SF, DCA separates the CFRs of the malicious CHs from those of normal nodes.

We improve the DCA, which depends on DBSCAN. Eps and Minpts are two important parameters in DBSCAN. Eps is the radius of a point’s neighborhood. When Eps is set, Minpts, the number of points, works as the density threshold. The following are some concepts and principles of the DBSCAN [36,37,38,39] with parameters (Eps, Minpts).

In DCA-SF, the data set (DS) consists of points in two-dimensional space, i.e., *DS* = {*x*_1_, *x*_2_, …, *x_m_*} where m is the number of CHs. When an SN receives the CFRs of CHs from INs in the *n*-th round, the value of *x_i_* can be arranged by Equation (10). The calculation of CH_i_s CFR in the *n*-th round is given in Section 3.2. Setting *x_i_* in this way can enhance the stability of data clustering results in successive rounds.
(10)$$x_{i}\;=\;\left(CFR\_CH_{i}\left(n\right),CFR\_CH_{i}\left(n-1\right)\right)$$Eps-neighborhood of a point: The Eps-neighborhood of a point *x_i_* is defined by
(11)$$N_{Eps}\left(x_{i}\right)=\{x_{j}\in DS|dist\left(x_{i},x_{j}\right)\leq Eps\}$$
where *dist*(*x_i_*, *x_j_*) is the distance between *x_i_* and *x_j_*, and *N_Eps_*(*x_i_*) is the Eps-neighborhood of *x_i_*.Directly density-reachable: A point *x_j_* is density-reachable from a point *x_i_* if (1) *x_j_* ∈ *N_Eps_* (*x_i_*) and (2) *N_Eps_* (*x_i_*) ≥ *Minpts*. If a point satisfies condition (2), this point is a core point.Density-reachable: If ∃ a chain of points *x*_1_, *x*_2_, …, *x_c_* ∈ *DS*, where any two successive points are directly density-reachable, then the points *x*_1_ and *x_c_* are density-reachable. If a point’s density is not a core point but it is density-reachable from a core point, then this point is a border point.Density-connected: A point *x_j_* is density-connected to a point *x_i_* if there is a point *y* ∈ *D*, where both *x_j_* and *x_i_* are density-reachable from *y*.Data cluster: A data cluster (DC) is a nonempty subset of DS satisfying:
(1)∀ *x_i_*,*x_j_*, if *x_i_* ∈ *DC* and *x_j_* is density-reachable from *x_i_*, then *x_j_* ∈ *DC*.(2)∀ *x_i_*,*x_j_* ∈ *DC*, *x_i_* is density-connected to *x_j_*.Noise: Let DC_1_, DC_2_, …, DC_k_ be data clusters of the data set DS. Noise is the point not belonging to any data cluster DC_i_ in the data set DS, i.e., *noise* = {*p* ∈ *DS*|*∀ i: p* ∉ *DC_i_*} *i* = 1, 2, …, *k*.

DBSCAN divides data points into different data clusters based on density. The judgment is whether the number of points in the circle Eps-neighborhood is larger than or equal to Minpts. If there are at least Minpts points inside a point’s Eps-neighborhood, the point is called a core point. If there are less than Minpts points inside a point’s Eps-neighborhood, and this point is inside a core point’s Eps-neighborhood, then it is called a border point. Other points are called noise points. The pseudo-code of the algorithm is given below.
Input: sample data set DS, parameters (Eps, Minpts) Output: data cluster set C DBSCAN (DS, Eps, Minpts) Begin
Mark all points in DS as unvisited;DoRandomly choose an unvisited *x_i_*;Mark *x_i_* as visited;If points in *x_i_*’s Eps-neighborhood are no less than Minpts Create a new data cluster (DC); Set *N* consist of points in *x_i_*’s Eps-neighborhood;  For each point *x_j_* in *N*   If *x_j_* is unvisited    Mark *x_j_* as visited;    If points in *x_j_*’s Eps-neighborhood are no less than Minpts, add points to N;    End if    If *x_j_* is not a member of any data cluster, add *x_j_* to DC;    End if   End if  End for;  Output DC;Else mark *x_i_* as a noise point;End ifUntil all unvisited points are visited;Output C
End

#### 3.3.2. Dynamic Parameter DBSCAN (DP-DBSCAN)

In our DCA-SF, the parameters (Eps, Minpts) change depending on the network scenarios. It is the dynamic parameters instead of the preset ones based on previous experience [36] that DP-DBSCAN employs to offer better reliability in changing situations. The methods of regulating these parameters adaptively in [40] are so complicated that they cannot be applied in a WSN. To apply the DP-DBSCAN to WSNs, we design a method to set Eps and Minpts, expecting a better performance in detecting a selectively forwarding attack.

The value of Eps is determined by the distribution of points in the data set. If the Eps is too large, a noise point would be judged as a normal one; if the Eps is too small, a normal point would be judged as a noise point. Based on the two-dimensional data set including *d*_1_ = (*x*_1_*, y*_1_), *d*_2_ = (*x*_2_, *y*_2_), …, *d_m_* = (*x_m_*, *y_m_*), the center point *d* = (*x’*, *y’*) is calculated by Equations (12) and (13). Then, the Eps is calculated by Equation (14), where *distance*(*p*, *q*) denotes the Euclidean distance between points *p* and *q*.
(12)$$x’=\frac{1}{m}*\sum_{i=1}^{m}x_{i}$$
(13)$$y’=\frac{1}{m}*\sum_{i=1}^{m}y_{i}$$
(14)$$Eps=\frac{1}{m}*\left(distance\left(d_{1},d\right)+distance\left(d_{2},d\right)+\ldots +distance\left(d_{m},d\right)\right)$$

*Minpts* is especially defined in Equation (15).
(15)Minpts=⎣(c/b)+0.5⎦
where *c* is the number of clusters in the WSN and *b* is an integer 1 < *b* < 12, which will be discussed in Section 4.2.2.

#### 3.3.3. Complexity Analysis of DP-DBSCAN

The time complexity of ordinary DBSCAN is O(m^2^) [41], where m is the number of samples in the data set that needs to be clustered. The work of [41,42,43] reduces the time complexity to O(m) or O(m*log (m)), respectively. In DCA-SF, the DBSCAN algorithm is not too complex for SN and INs.

To calculate the parameters (Eps, Minpts) for DBSCAN, 2(m–1) + m additions and three divisions are performed in Equations (12)–(14). In addition, m times distance calculations need to be performed. So, the time complexity of DP-DBSCAN is O(5m) = O(m). In the proposed scheme, m is the number of CHs when it performs in the SN or sensor nodes in one WSN cluster when it performs in INs.

### 3.4. Implementation of DCA-SF

#### 3.4.1. Two Key Notes

One key point in DCA-SF is that the CFRs of nodes are employed. Table 1 and Table 2 give an intuitive grasp of the advantage. In a WSN, because the CFR is more stable than the SFR (single round forwarding rate), DCA-SF can pick malicious CHs out by their CFRs.

The other key point in DCA-SF is that the SN charges any CH as a suspect only when this CH is detected as abnormal in the successive *k* rounds. Now, the detection enters the called stable status. The earlier the stable status occurs, the faster the detection. After that, the suspected CH will be judged as malicious or innocent by the SN according to the report from its IN.

#### 3.4.2. Process of DCA-SF

The DCA-SF scheme consists of centralized and distributed schemes. In the centralized detection scheme, the SN executes DP-DBSCAN independently after receiving CFRs of all the CHs from INs. Conducting centralized testing takes place for two reasons. On one hand, the SN is strong enough to implement the DP-DBSCAN without caring for the algorithm’s time complexity and space complexity. On the other hand, the SN can never become malicious as assumed at the beginning of Section 3; thus, the results of clustering CH CFRs by the SN are highly reliable. If the CFR of a CH is determined as a noise by the SN, it is anomalous to other CHs’ CFRs. Only if the CFR_CH_i_ is judged as noise in *k* successive rounds can CH_i_ be marked as a suspect node. The result may come from a malicious node or a poor channel. IN_i_ performs DP-DBSCAN on the CFRs of CH_i_ and MNs in cluster *i* to confirm whether the CH_i_ is malicious or not. This processing is called distributed detection.

If *k* is too small, the MDR will be low. However, it will increase the number of times one CH is marked as suspect, which will increase the number of times that IN performs detection. The computational energy consumption of IN increases. If *k* is too large, the FDR will be low. However, a malicious CH that should be marked as the suspect may exist in the network for a long time, increasing the number of dropped packets. The value of *k* is set in advance and is fixed during network running. So, the value of *k* depends on the user’s needs for the detection results. If the user requires a lower MDR, the value of *k* can be set smaller. If the user requires a lower FDR, the value of *k* can be set larger. In our simulation, the value of *k* is set to 4, 5 and 6. Figure 4 shows the detection process of the proposal scheme DCA-SF, where *IN_i_* and *CH_i_* represent the IN and CH of cluster *i*, respectively.

## 4. Simulation Results and Analysis

### 4.1. Simulation Parameters and Data Set

#### 4.1.1. Simulation Parameters

The simulation parameters in MATLAB 2014a are shown in Table 3.

The function rand (1, 1) in Table 3 is to generate numbers subjected to randomly uniform in (0, 1). The LEACH clustering method is used in our simulations; the selecting of CH and the forming of the cluster are random. In simulations, we define two metrics (MDR, FDR) as follows. The missed detection means that a malicious node is regarded as normal; the missed detection rate (MDR) is the ratio of missed detection CHs to the total malicious CHs. The false detection means that a normal node is regarded as malicious; the false detection rate (FDR) is the ratio of false detection CHs to the total CHs.

According to the results of DP-DBSCAN, the point will be judged as noise if it is clustered as class 0.

#### 4.1.2. Data Set

As shown in Table 3, in simulation, the number of malicious CHs is the total number of CHs x the ratio of malicious CHs. It is subjected to a random uniform distribution. Then, an id set (IS) of malicious CHs is formed randomly. If the id of a CH belongs to IS, this CH will be malicious. We assume the total forwarding rates of malicious CHs (MNs) and normal CHs (MNs) obey a random uniform distribution in [0, 0.7] and [0.8, 1], respectively. The latter depends on channel quality. The dropped packets by CHs can be counted out. So, the number and rounds of malicious dropping packets are random, and the data set will be formed in every round by calculating the CFRs of CHs and MNs with Equations (7) and (9).

### 4.2. Simulation Results and Analysis

#### 4.2.1. An Example of Simulation

In our simulations, the CHs’ distribution according to the LEACH algorithm is identical to that in Figure 5. As shown in Figure 5, the network has been divided into 15 clusters in this simulation, and each cluster has selected a CH and an IN.

The total forwarding rates produce randomly as listed in Table 4. In this simulation, the total forwarding rates of normal CHs are located at [0.8052, 0.9402]. CH_2_, CH_8_, and CH_12_ are malicious nodes. The total forwarding rates of these three malicious nodes are 0.6299, 0.1468, and 0.6051 respectively, and assume that *k* described in Section 3.4.2 is set to five in this simulation.

The classification of each CH after DP-DBSCAN clustering in each round is shown in Table 5. The CFR of CH_8_ is clustered as zero classifications from the first round. This shows that CH_8_ has dropped many packets from the first round. Different from CH_8_, the total forwarding rates of CH_2_ and CH_12_ are closer to normal; their CFRs are clustered as class 1 in the first five rounds. However, as the network runs, their CFRs gradually show differences. So, they are clustered as zero after the fifth round. The detection results become stable after the fifth round. The SN broadcasts the message (CH_2_, CH_8_, and CH_12_ are regarded as suspected) to IN_2_, IN_8_, and IN_12_ respectively. After IN_2_, IN_8_, and IN_12_ have done DP-DBSCAN in their clusters, they are confirmed as malicious. So, the MDR and FDR in this simulation are both zero.

#### 4.2.2. Results and Analysis

To make it clear how Minpts affects the detection result, we perform 500 simulations and get the statistical results. In the 500 simulations, the ratios of malicious node are Ar = 5%, Ar = 10%, Ar = 15%, Ar = 20%, and Ar = 25%. In these 500 simulations, *k* described in Section 3.4.2 is set to five. In every ratio, we change the value of b in Equation (15) to conduct 100 simulations. Then, we get the relationships between MDR and b, FDR and b, and the number of rounds at which the detection results becoming stable and b. Figure 6 shows the relationship between MDR and b in different malicious node ratios.

In Figure 6, the curve representing Ar = 5% is covered by the curve representing Ar = 10%. The MDR is 0 in both situations where Ar = 10% and Ar = 5%. Minpts changes with the number of CHs in WSNs. The average MDR under different malicious nodes is counted out by 10 simulations in the same ratio and b. The overall trend is that the average MDR increases as the b value increases.

In Figure 7, the curve representing Ar = 20% is covered by the curve representing Ar = 25%. FDR is 0 in the both situations of Ar = 20% and Ar = 25%. The relationship between the average FDR and b is shown in Figure 7, in which the average FDR reduces as b increases.

The relationship between average stable rounds and b is shown in Figure 8.

The average rounds shown in Figure 8 are counted to reflect the detection speed. The fewer rounds required for the detection results to stabilize, the higher the detecting speed. The overall trend shown in Figure 8 is not obvious, and an obvious overall trend will be shown in the following.

In Figure 9, the overall trend is obviously shown. The average MDR increases as the b value increases. The average MDR of *k* = 6 is higher than that of *k* = 5, and so *k* = 5 is set to *k* = 4.

In Figure 10, the average FDR decreases as the b value increases; the average FDR of *k* = 4 is higher than that of *k* = 5, and that of *k* = 5 is higher than that of *k* = 6.

Figure 11 shows the average number of rounds at which the detection results become stable for all the simulations for different values of b in situations of *k* = 4, *k* = 5, and *k* = 6.

According to the statistical results, when b = 3 and *k* = 5, the average MDR of DCA-SF is 1.04%, the average FDR of DCA-SF is 0.42%, and the detection results become stable from an average of 5.5 rounds. 

Furthermore, the value of b may change as the application scenario changes. If users focus more on the lower MDR, the value of b can be taken as smaller; if users focus more on the lower FDR, the value of b can be taken as larger.

#### 4.2.3. Results Analysis

As shown in Figure 6 and Figure 9, the average MDR increases as the b value increases. This conclusion means that when Eps and c are constant, the average MDR increases as the Minpts decreases. This is because when the neighborhood radius is fixed, decreasing the value of Minpts will make the point with fewer points in its neighborhood not be judged as noise. All points will not be judged as noisy if Minpts reduces to zero. This will lead our scheme to mark all the CFRs of malicious nodes as normal. With the same reason, all the CFRs of normal nodes are not marked as anomalous. So, the average FDR reduces as the Minpts decreases; the results are shown in Figure 7 and Figure 10.

As shown in Figure 11, the average rounds at which the detection results become stable is almost stable as b changes because this average number is determined by the manner of dropping packets. In our simulation, the number and rounds of malicious nodes dropping packets are random, so the time at which the detection results begin to stabilize is little correlated to Minpts.

The results show that small *k* values usually make MDRs better while large *k* values bring low FDRs. Users can set *k* according to their requirements.

### 4.3. Comparison to Other Schemes

#### 4.3.1. Detection Results Compared with Other DCAs

In this section, we compare the clustering results of DP-DBSCAN in DCA-SF with those of K-means in terms of the MDR and FDR. When K = 2, most of the CFRs in one cluster are normal, while the CFRs in the other cluster are anomalous. When K = 3, three clusters are normal, anomalous, and suspected, respectively. A total of 150 simulations are performed to compare the performance in different DCAs; in these 150 simulations, we set b = 3, *k* = 5.

As shown in Figure 12, in terms of the MDR, the performance of DP-DBSCAN is better than K-means (K = 2) and K-means (K = 3), and the time complexities of these three DCAs are all O(m).

The comparison of FDRs in different DCAs is shown in Figure 13. The results of the FDR in DP-DBSCAN are the same as those in K-means when K = 2, and the performances of them are better than those when K = 3.

As shown in Figure 12 and Figure 13, at the same time as the complexity O(m), the detection results of DP-DBSCAN are the best in these three DCAs. If the *K* value of K-means can be determined by some optimization algorithms, it can get a better result, but the time complexity and space complexity are too high to be performed in sensor nodes.

#### 4.3.2. Detection Results Compared with Other Schemes

We compare DCA-SF with the watchdog mechanism [15], neighbor-based scheme [13], and IN-based scheme [8] on two metrics of MDR and FDR. The behaviors of these four schemes are recorded in Table 6. The simulations on all four schemes are on the same scenarios.

#### 4.3.3. Energy Consumption Compared with Other Schemes

All nodes take part in detecting selective forwarding attacks directly in the watchdog mechanism [15] and the neighbor-based scheme [13], despite the cost of energy. In other words, these two schemes are far from energy-efficienct. So, we only compare our scheme with IN monitoring [8] in the metric of energy consumption.

Figure 14 shows an ideal deployment of sensor nodes in a WSN. More nodes lay near the SN (at the center) because they have to forward more data packets from nodes far from the SN in the multi-hop WSNs.

Figure 15 is the simulation result with the energy consumption module [44] in wireless communication.

DCA-SF offers the WSN a much longer lifetime than the IN monitoring scheme. Under the same simulation settings, the death of node occurs firstly in the 1210th round with DCA-SF, which is nearly 700 rounds later than when it occurs with IN monitoring. The fewer nodes involved, the less energy consumed in detection. Nodes taking turns as CHs and INs also helps balance the energy over the WSN, which contributes to the long lifetime of the network.

## 5. Conclusion and Future Work

To detect selective forwarding attacks in WSNs, we have proposed the DCA-SF scheme based on the DBSCAN. The DP-DBSCAN, based on DBSCAN, can cluster the abnormal behaviors of malicious nodes to improve the scheme’s intelligence. One is the radius of neighborhood Eps in DCA-SF determined by the distribution of points in the data set, and the other is Minpts in DP-DBSCAN, which is dependent on the number of points in the data set. When b = 3, *k* = 5, the statistical results show that the DCA-SF has an MDR of 1.04% and an FDR of 0.42%. The detection results become stable after an average of 5.5 rounds.

In the future, we will take a field test of this detection scheme on our WSN platform. Then, we will apply DP-DBSCAN to distributed WSNs. We plan to employ DPC and SNN-DPC to cluster the data set in SN for testing their effects.

## Figures and Tables

**Figure 1 sensors-20-00023-f001:**
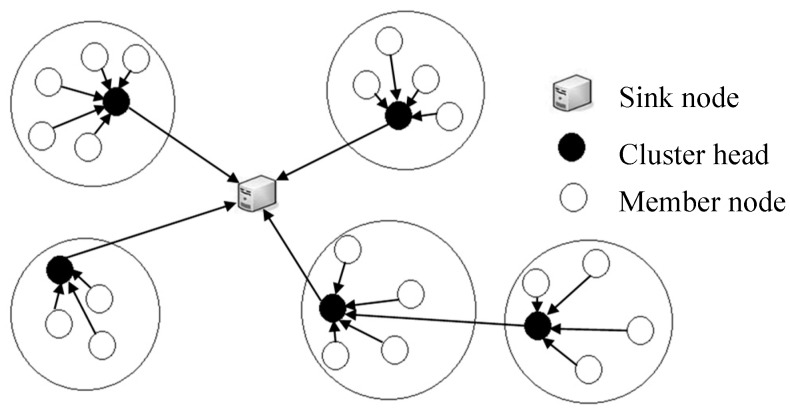
An example of a cluster-based wireless sensor network (WSN).

**Figure 2 sensors-20-00023-f002:**
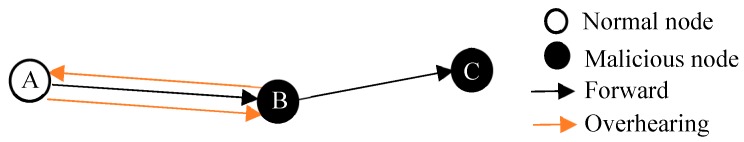
Watchdog monitoring mechanism.

**Figure 3 sensors-20-00023-f003:**
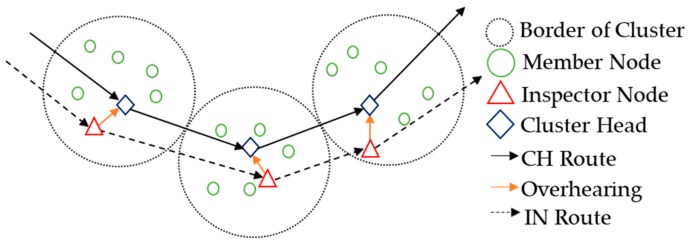
Detection frame.

**Figure 4 sensors-20-00023-f004:**
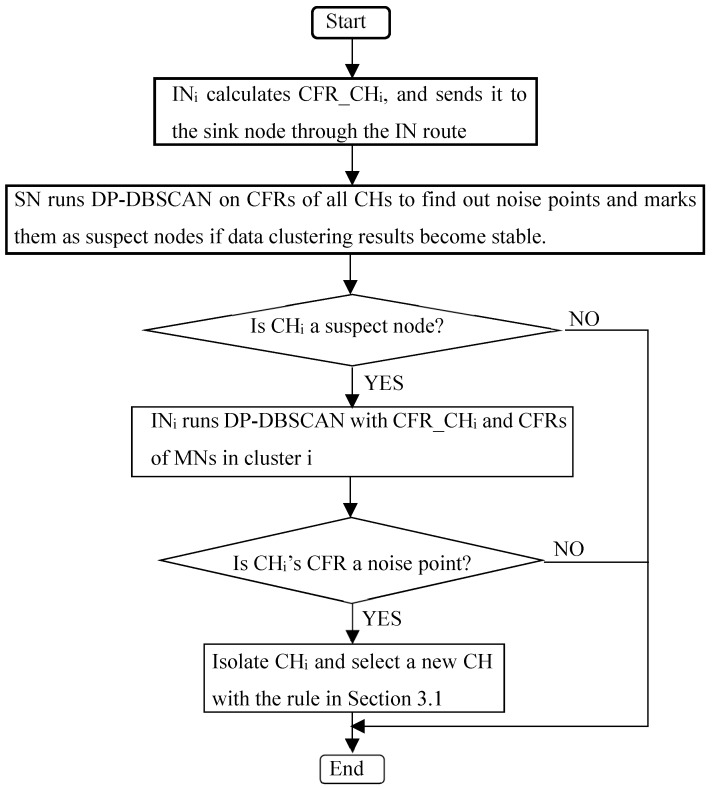
Detection flowchart.

**Figure 5 sensors-20-00023-f005:**
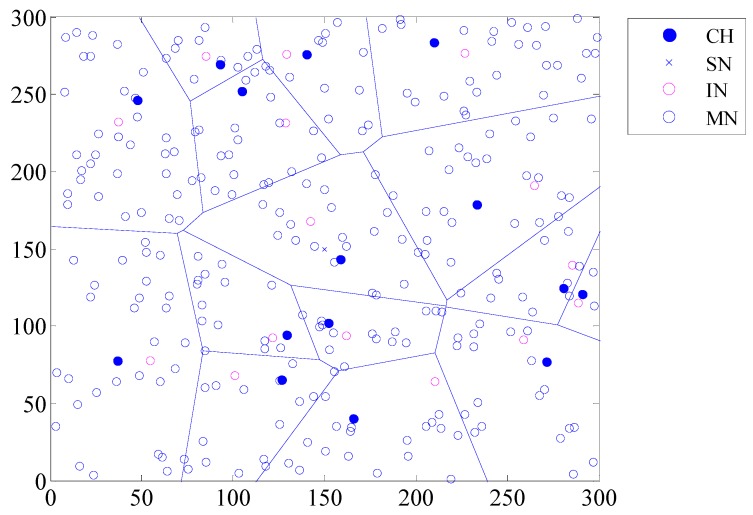
Network nodes distribution.

**Figure 6 sensors-20-00023-f006:**
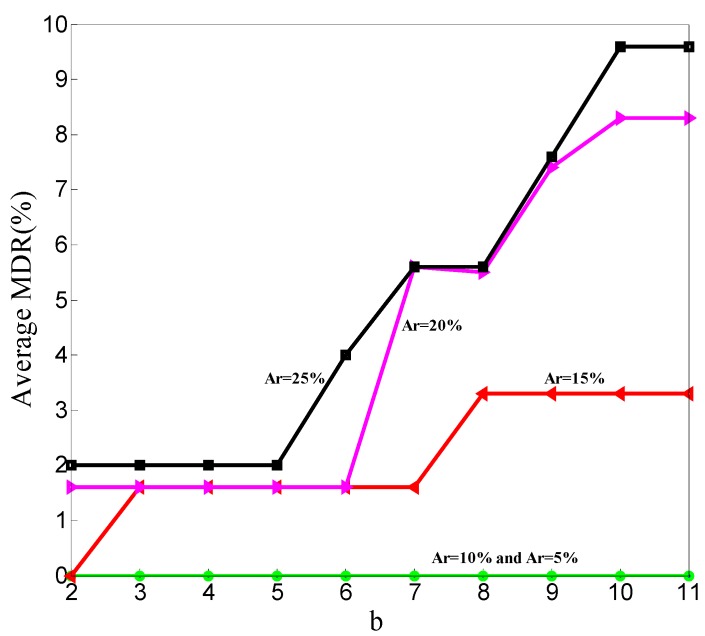
Relationship between average missed detection rate (MDR) and b.

**Figure 7 sensors-20-00023-f007:**
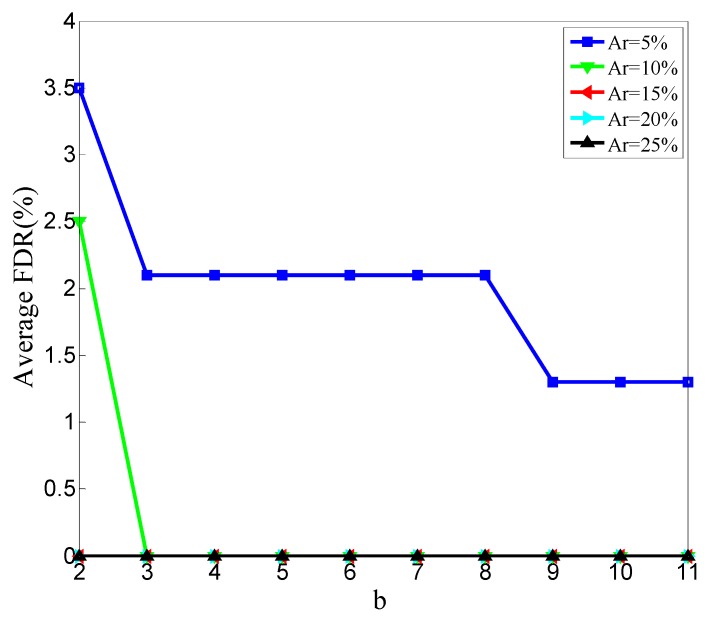
Relationship between average false detection rate (FDR) and b.

**Figure 8 sensors-20-00023-f008:**
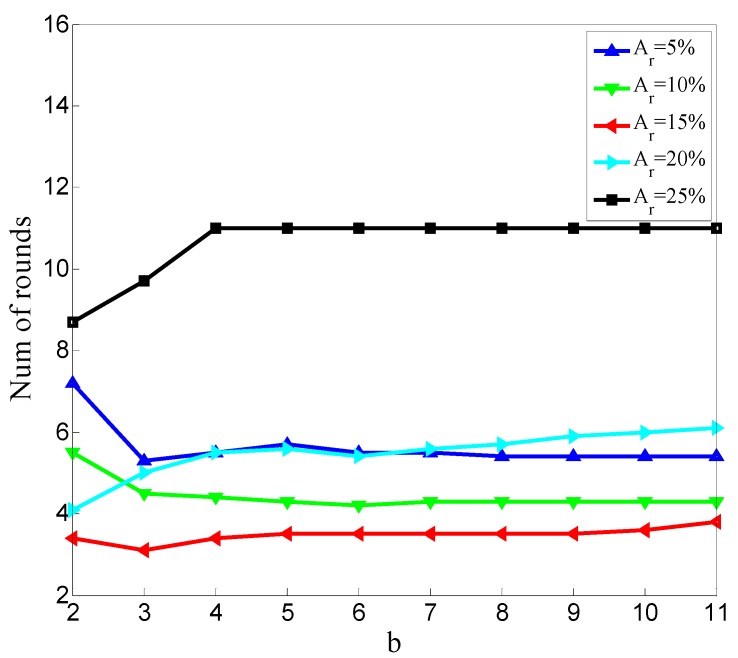
Relationship between the average stable rounds at which the detection results begin to stabilize and b.

**Figure 9 sensors-20-00023-f009:**
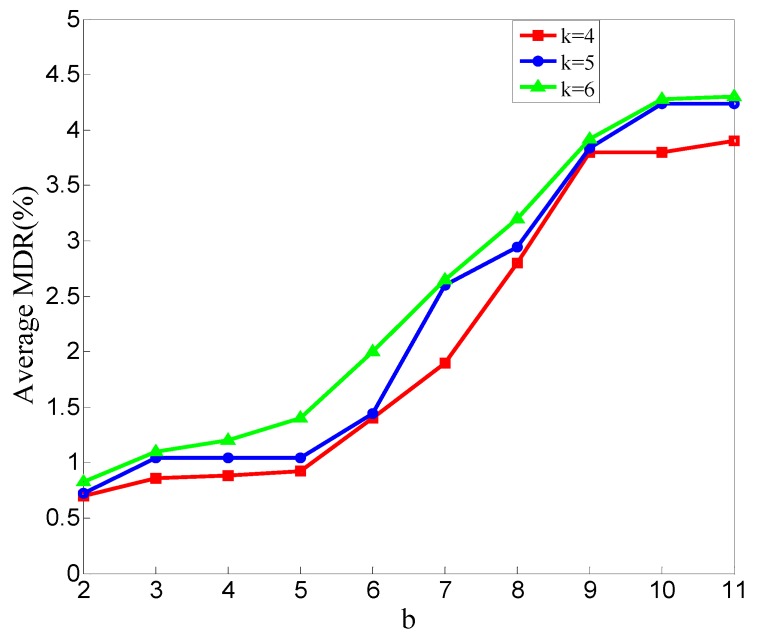
Average missed detection rate (MDR) of all experiments in different values of b.

**Figure 10 sensors-20-00023-f010:**
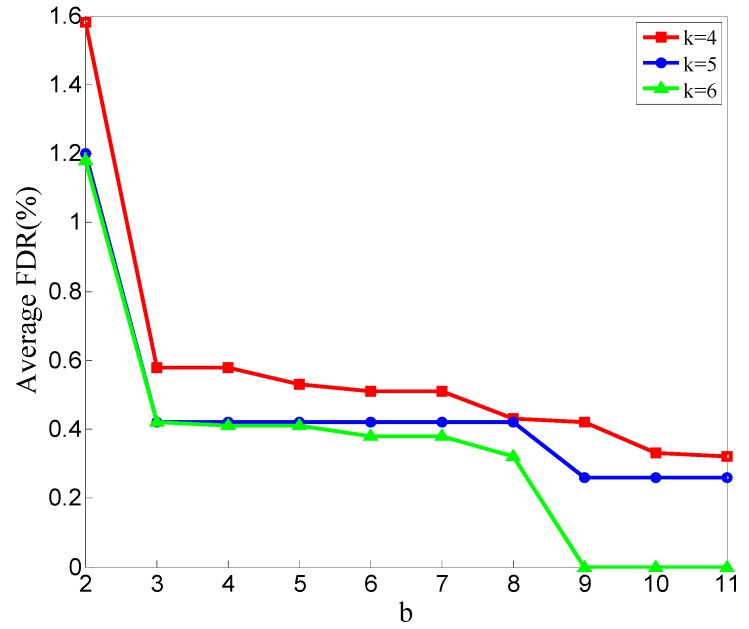
Average FDR of all experiments in different values of b.

**Figure 11 sensors-20-00023-f011:**
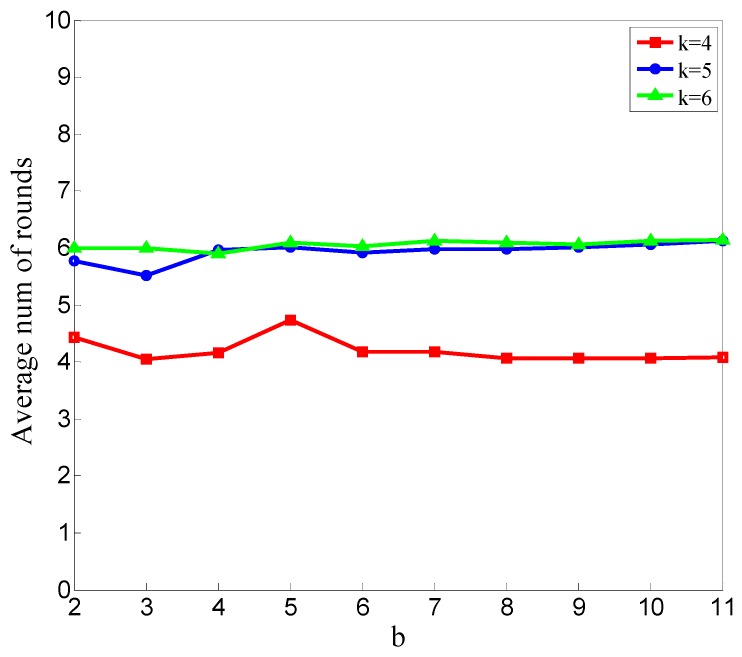
Average stable rounds at which the detection results begin to stabilize for different values of b.

**Figure 12 sensors-20-00023-f012:**
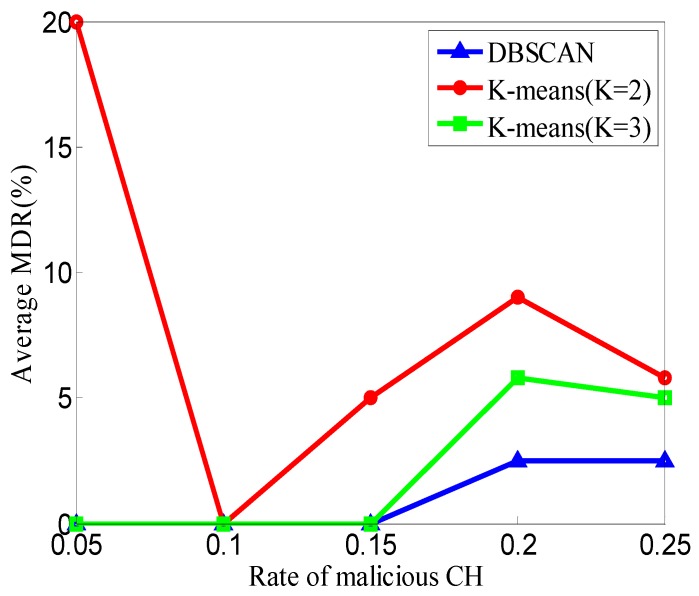
Comparison of MDR in different data clustering algorithms (DCAs).

**Figure 13 sensors-20-00023-f013:**
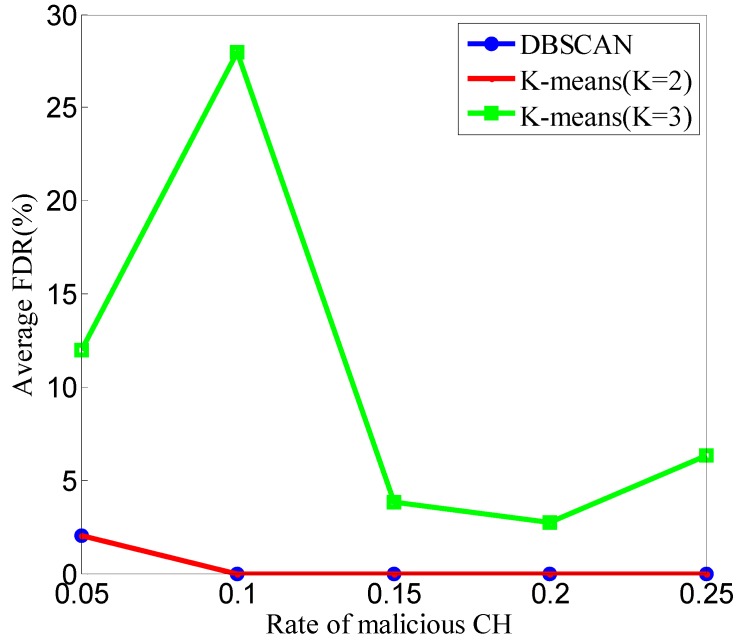
Comparison of FDR in different DCAs.

**Figure 14 sensors-20-00023-f014:**
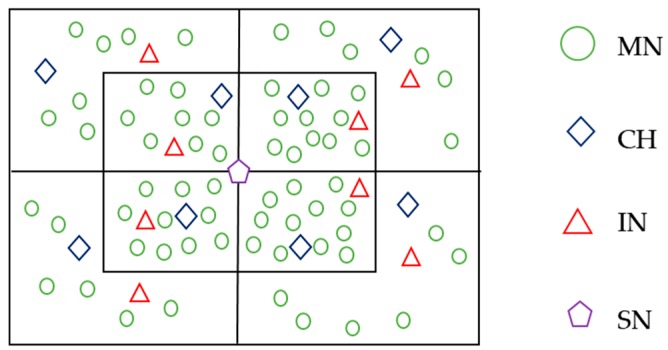
The best distribution of nodes in view of the energy balance.

**Figure 15 sensors-20-00023-f015:**
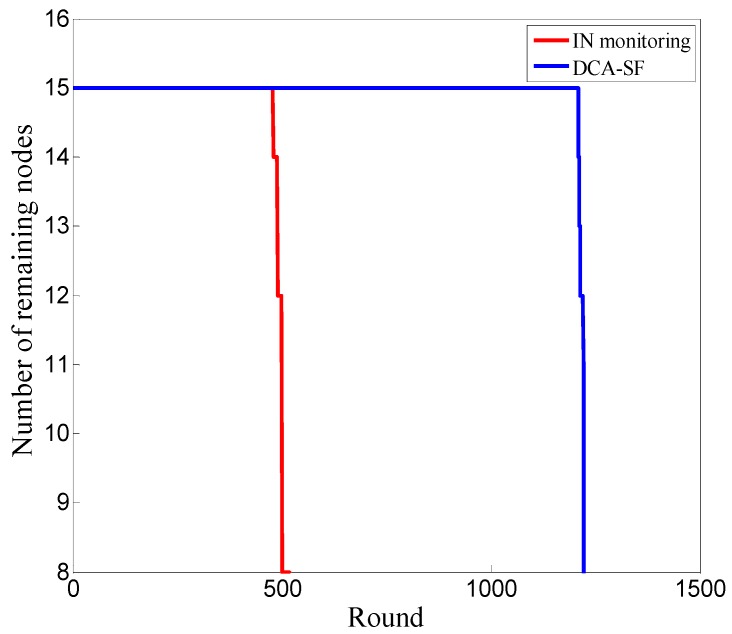
Comparison of network lifetime.

**Table 1 sensors-20-00023-t001:** Cumulative forwarding rates (CFR) of a cluster head (CH).

Round	1	2	3	4	5	6	7	8	9	10
**CFR**	0.9091	0.9545	0.9394	0.9318	0.9444	0.9375	0.9333	0.9186	0.8958	0.8962

**Table 2 sensors-20-00023-t002:** Single forwarding rate (FR) of a CH.

Round	1	2	3	4	5	6	7	8	9	10
**Single FR**	0.9231	1.0000	0.8463	0.7692	0.5385	1.0000	0.9231	0.9186	0.8463	0.9186

**Table 3 sensors-20-00023-t003:** Parameters of simulation. MN: member node, WSN: wireless sensor network.

Parameter Item	Parameter
Clustering method of WSN	LEACH
Area of the network	300 × 300
Total number of nodes	300
Ratio of malicious nodes	0.05 + 0.2 × rand(1, 1)
Total forwarding rate of normal nodes	[0.8, 1] (Change as channel quality changes)
Total forwarding rate of malicious node	0.7 × rand(1, 1)
Total forwarding rate of MN	[0.8, 1] (Change as channel quality changes)
The number and rounds of dropping packets	Random

**Table 4 sensors-20-00023-t004:** Total forwarding rate of each CH.

**No. CH**	**1**	**2**	**3**	**4**	**5**	**6**	**7**	**8**
**Forwarding rate**	0.8755	0.6299	0.9074	0.8057	0.8052	0.9402	0.9030	0.1468
**No. CH**	**9**	**10**	**11**	**12**	**13**	**14**	**15**	
**Forwarding rate**	0.8256	0.8851	0.9336	0.6051	0.9017	0.8862	0.8183	

**Table 5 sensors-20-00023-t005:** CFRs clustering results in each round.

	No. Round	1	2	3	4	5	6	7	8	9	10	…	49	50
No. CH														
**1**		1	1	1	1	1	1	1	1	1	1	…	1	1
**2**		1	1	1	1	1	0	0	0	0	0	…	0	0
**3**		1	1	1	1	1	1	1	1	1	1	…	1	1
**4**		1	1	1	1	1	1	1	1	1	1	…	1	1
**5**		1	1	1	1	1	1	1	1	1	1	…	1	1
**6**		1	1	1	1	1	1	1	1	1	1	…	1	1
**7**		1	1	1	1	1	1	1	1	1	1	…	1	1
**8**		0	0	0	0	0	0	0	0	0	0	…	0	0
**9**		1	1	1	1	1	1	1	1	1	1	…	1	1
**10**		1	1	1	1	1	1	1	1	1	1	…	1	1
**11**		1	1	1	1	1	1	1	1	1	1	…	1	1
**12**		1	1	1	1	1	0	0	0	0	0	…	0	0
**13**		1	1	1	1	1	1	1	1	1	1	…	1	1
**14**		1	1	1	1	1	1	1	1	1	1	…	1	1
**15**		1	1	1	1	1	1	1	1	1	1	…	1	1

**Table 6 sensors-20-00023-t006:** Comparison of four schemes.

	Metric	Ratio of Malicious Node	Number of Simulations	MDR	FDR
Scheme					
**Watchdog** [15]		10%	100	4%	28.2%
**Neighbor-based monitoring** [13]		10%	100	3%	17.2%
**IN monitoring** [8]		10%	100	1%	2.5%
**DCA-SF**		10%	100	0	0

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
