# Peer review of "A Data Clustering Algorithm for Detecting Selective Forwarding Attack in Cluster-Based Wireless Sensor Networks"

_sensors, 2019, doi:10.3390/s20010023_

Round 1

Reviewer 1 Report

After reading the whole paper I have a clear idea of authors' purposes and their approach and it seems to me it is quite interesting. But I have the following remarks:

In section 2 authors mention just 2 Data Clustering Algorithms: K-means and K-medoids; but there are other based on hierarchy, or density that are not properly discarded. It is true that DPC has become more interesting since 2014, but only mentioning 2 DCA seems too poor. In other literature (https://doi.org/10.1016/j.ins.2018.03.031), some DBSCAN drawbacks are mentioned, but in this paper author don't explain why that is not meaningful for the targeted scenarios. I suggest changing the order of main concepts in section 2. I suggest talking first about schemes against selective forwarding attack and DCA afterwards, because in section 2.2 authors presents the use of DCA as a countermeasure for selective forwarding attack and it provides the perfect excuse for talking afterwards of DCA (present section 2.1) In paragraphs from line 254 to 271 authors describe how Eps and Minpts are estimated. But it is not clear to me. First, Eps and Minpts are changing parameters depending on the network scenarios. But, do they change along time or you just define them for a specific deployment and they remain the same if deployment doesn't change? WSN deployment changes continuously as nodes die. It seems that adapting those parameters in run time is very expensive from a computational point of view. Authors then provide an affordable way to readily estimating them in a WSN. But, where is the rationale for those equations? Are justified in scientific literature? May be the rationale is in the paper, but I didn't realize. Authors mention that DP-DBSCAN is very expensive computationally for WSN nodes and thus, SN runs DP-DBSCAN for the data provided by the IN. If SN suspects of a malicious CH, IN performs DP-DBSCAN to discard poor cahnnel quality. But IN is actually a WSN node with low computational resources. Why in line 303 you select 5 rounds to be sure a suspicious CH is really a malicious CH? By 5 and not 6? I guess that sentence "… them as suspect nodes until data clustering results" in Figure 4 is incomplete. In figure 6, for Ar = 10%, MDR is 0%, no matters b value. So, for a 5% of malicious nodes compared to the total of nodes MDR increases with b. For Ar values of 15%, 20% and 25% increases also; but for 10% is always 0. Why? Why those results didn't catch authors attention? In the introduction authors should also mention that WSN node have low computational resources. They mention in later sections. In future works, I guess that authors should consider running a field test, because real and simulated WNS behaviours are different. English spelling check: In line 101 I suggest "This algorithm is based on …" Equation 3 is wrong, I am afraid. Check equation 4 in https://doi.org/10.1016/j.ins.2018.03.031 Line 173: "…dangerous attacks are from…" Line 183: "…data packets consisting on environment information…" Line 208: 'which' instead of 'with'. Line 301: Letter 'S' at the end means 'SN?, I guess. Line 319: "… a malicious node is regarded as…" Line 407: "…stable is little correlated to Minpts."

Author Response

Thanks for your comments. Your suggestions are very helpful for improving the research method and description and then promoting the paper quality greatly. Appreciate your help that you have pointed out grammar errors so patiently! The errors have been corrected in the revised manuscript.  The reply to your review report has been submitted.

Reviewer 2 Report

The paper deals with the interesting topic of hardening a wireless sensor network against selective forwarding attack and protecting the energy resources of each node of such a network, thereby extending the life of such a network.

I have some insights and comments. To help the reader and to improve
the quality of the manuscript I suggest to modify/consider the
following aspects:
l.30: It is true that the rest of the paper shows that each CH node does not need to be able to directly exchange data with the SN node, but it can use other CH nodes for this purpose. It seems to me that at the beginning of the description it is worth writing about it.

l.58: The rule according to which the full meaning of the abbreviation should be given before using the abbreviation must be followed. In the content of paper on line 58, the abbreviation CFR is used for the first time (in this case it should not be taken into account that the definition of this abbreviation is given in the abstract - the abstract often occurs regardless of the paper).

l.63: Here, the definition of "cumulative forwarding rate (CFR)" will not be needed when the abbreviation CFR is defined on line 58.

l.67: There is no definition of DCA in paper, but this definition is in the abstract.

l.105: Formula (1): At this point, there is no information on where d_c parameter comes from and what its purpose is.  The question arises: can d_c be greater than d_ij?  It is worth clearer to describe it in the paragraph beginning in line 107, and in particular to write that the function chi(x) has value one when the distance between two nodes is smaller than the set value of d_c.

l.108: By what formula is delta_i calculated? From formula (2) or from formula (3)? Or maybe formula (3) describes the property of a certain node?

l.181: In Figure 3, the nodes are marked with small circles. The color of the circle determines the role of the node. If the paper is printed on a black and white printer, the difference becomes invisible. Maybe it is worth filling individual circles with different textures?

l.199-206: Does "Forw_MN" appearing on the left of formulas (5) and (7) mean the same as "ForwMN" (MN is lower index) appearing on the right in these formulas? The same question applies to "Reci_CH" in formula (4).
The parameter names used in formulas (4)-(8) make these formulas difficult to understand. A better solution is probably to use, for example, FORW_CH, FORWCH (CH is lower index) and forwCH (CH is lower index), of course, if these are different designations.

l.233: First use of "Minpts". It is not known what this means?

l.243: Redundant "."

l.245: What actions in the procedure are envisaged for nodes belonging to the "noise" set?

l.274-275: The sentence starting with the words: "After improving, according to the schemes in [36-38], it is easy to improve its time complexity ..." is very intriguing.

l.280: The title of section 3.4 should be transferred to a new page.

l.328: The paper uses two very similar abbreviations that definitely hamper the understanding of the content:
MN - malicious node
NM - number of malicious nodes
I suggest you opt out of the abbreviation NM and use the full name.

l.396: The title of section 4.2.3 should be transferred to a new page.

l.406: The word "and" seems unnecessary.

Author Response

(The authors gave the same response as above.)

Round 2

Reviewer 1 Report

All my suggestions have been included in a very proper way.

I recommend to spell check the paper just in case. In page 4, you have repeated references in the same line for K-means and K-medoids.

No extra suggestions. I hope you test your algorithm in a real field test and you can verify your analysis. Your weak spot is proposing a complex algorithm in a node (IN), even though it is run from time to time.

Author Response

Thanks for your suggestions. They do contribute to promoting the paper quality a lot.

This manuscript is a resubmission of an earlier submission. The following is a list of the peer review reports and author responses from that submission.

Round 1

Reviewer 1 Report

in thsi paper, the author used clustering algorithm such as DBSCAN to dectect selective-forwarding attack in WSN. The simulation results show the proposed method is promissing.

However, there are some aspects need to improve as bleow.
1. the basic concepts of DBSCAN are revisited, but the expresssions are not authentic, please refer to the original definitions in DBSCAN.
e.g, "ϵ-neighborhood: When ?? ∈ D ,...." --> "(Eps-neighborhood of a point) The Eps-neighborhood of a point p, denoted by N_Eps(p) is defined by ...."

2. A long blank in page 6.

3. [24] is not precise DBSCAN, by the way, I am not sure it runs in O(nlog(n)) expected time, but \rho-approximate DBSCAN ("DBSCAN Revisited: Mis-Claim, Un-Fixability, and Approximation, 2015 sigmod") runs in O(n) in low dimension, and NQDBSCAN ("A fast clustering algorithm based on pruning unnecessary distance computations in dbscan for high-dimensional data, Pattern Recognition 83 (2018) 375–387.") runs in O(nlog(n)) , please cite them.

4. figures are all too big.

5. why are the mails of the aurthors 163. com or qq.com ? not the official mails of the institudes that the authors work for?

6. Fig 10 is in page 17, while the caption is in page 18.

7. I didn't find any description of experimental data sets

8. the complexity of DP-DBSCAN is not provided.

9. in equ (6) and (7), the selection of [eps, minpts] is simple, is there any advantanges do this way? give us more explanations. There are many other works for automaticlly determining [eps, minpts] such as
[1] Karami, Amin, and Ronnie Johansson. "Choosing dbscan parameters automatically using differential evolution." International Journal of Computer Applications 91.7 (2014): 1-11.

the paper doesn't discuss the related works that involve in how to choose parameters.

Author Response

Thank you for your question and advice on polishing up our manuscript for promotion.Your comments are very helpful for us.

Reviewer 2 Report

First of all, the paper needs some proofreading to check grammar errors, typos, and improve readability.

The motivation and the research question are also unclear. For example, what is the primary goal of the paper? What are the contributions to the state-of-the-art? This should be emphasized in the introduction. 

Authors need to justify some of their choices. For example, the parameters of the simulation. Why were such parameters chosen? Is there a related work that helps understand it? 

The paper also lacks an in-depth discussion of the results. For the proposed scenario, MDR and FDR metrics were both 0. Why? In Machine Learning, it is common to use metrics such as precision, recall, f1, AUC, and so on. Why are such metrics not employed here? The other three schemes compared to the DCA-SF can be considered state-of-the-art for the selective forwarding attack problem? This is unclear in the manuscript. 

First, authors should present a research gap. For example, the existing schemes have a problem with energy consumption, or some metrics (precision or recall) does not achieve good results. Using this motivation, the authors propose a method and then compare it to the state-of-the-art. This research process was not done adequately in the paper. 

Other issues:

Table 4 - font size is too small

Reference 11 is not appropriately cited in the text (Das S or Semanti?)

Energy consumption is compared only with the IN-monitoring scheme. Why?

Author Response

Thank you for your question and advice on polishing up our manuscript for promotion.Your comments are very helpful.

Reviewer 3 Report

The paper describes a method to detect selective-forwarding attacks using DBSCAN based clustering algorithm.

The proposed methodology is somewhat hard to evaluate because the results and analysis are merely based on simulations where forwarding rates are controlled. It would have been better if some real implements or test environment was created to evaluate and explain the steps of the algorithm with respect to that.

The authors need to clearly identify the contribution of this paper to the scientific field. From the paper it is not clear, DBSCAN algorithm is not new, its a known clustering algorithm. Also, there are other clustering algorithms. It's not clear why the authors chose this specific algorithm and what advantages has to this particular application.

The paper is generally well-written. But at a few places, it fails with sentences either have typos or grammatical mistakes. Making it harder to understand the intent of the content.
- 3.4.1 Thinking of Proposed Method
- Page 2 Paragraph 2

The statement "Most schemes do not take into account the impact of channel quality on the cumulative forwarding rate (CFR)." - Either references are missing or qualitative analysis is required.

"After receiving the CFRs of all CHs from INs, the sink node starts to perform DP-DBSCAN algorithm in the data set."
-- This statement is confusing. "perform DP-DBSCAN algorithm in the data set" - What is the dataset in this case? Is the algorithm in Dataset?

If DP-DBSCAN is performed in INs and CFRs are sent from CHs. What if CH is malicious and sends a predetermined CFR? Would it be possible and if possible, how can it be detected in this approach.

Figure 5 is hard to understand.

4.3.1 Detection Results Compared with Other Schemes - Needs to properly describe the testbed and if all schemes are subjected to the same test. Also, need to explain the derivation of Table 6 results.

The authors need to resolve and clarify the above questions and sections. Additionally, the detailed steps and procedure on how their proposed algorithm is incorporated into existing WSNs need to write more clearly.

Author Response

(The authors gave the same response as above.)

Round 2

Reviewer 1 Report

Since clustering and WSN are two important parts of this work, the related work section 2, in my opinion, is better to be divided into two subsection, e.g. 2.1 WSN and 2.2 Clustering.

Only DBSCAN is introduced, but I think that each density-based clustering algorithm is suitable in this field, such as Density Peak, meanshift, as well as their improved versions, FastDPeak, EDDPC, LSH-DDP. such as:

[1]Rodriguez and A. Laio, "Clustering by fast search and find of density peaks," Science, vol. 344, pp. 1492-1496, 2014.

[2]Ankerst, M. M. Breunig, H.-P. Kriegel, and J. Sander, "OPTICS: ordering points to identify the clustering structure," in ACM Sigmod record, 1999, pp. 49-60.

[3] Y. Cheng, "Mean shift, mode seeking, and clustering," IEEE transactions on pattern analysis and machine intelligence, vol. 17, pp. 790-799, 1995.

[4]S. Gong, Y. Zhang, EDDPC: An efficient distributed density peaks clustering algorithm, J. Comput. Res. Dev. 53 (6) (2016) 1400–1409. 

[5] Y. Zhang, S. Cheny, G. Yu, Efficient distributed density peaks for clustering large data sets in mapreduce, in: IEEE International Conference on Data Engineering, 2017, pp. 67–68.

[6]Y Chen, X Hu,  W Fan, Fast Density Peak Clustering For Large Scale Data Based On kNN. Knowledge-based System. 2019. https://doi.org/10.1016/j.knosys.2019.06.032. 

Some explanations should be given that why we use density-based clustering here. The sentence "If intelligent detection schemes If intelligent detection schemes deriving from machine learning algorithm are applied in selective are applied in selective forwarding attack in WSN, it will be good for detecting cunning attack patterns" is feeble.

Reviewer 2 Report

The authors provided a revised version of the manuscript with some improvements. However, the paper still has some issues regarding the research design and presentation/discussion of results.